# PlWRKY13: A Transcription Factor Involved in Abiotic and Biotic Stress Responses in *Paeonia lactiflora*

**DOI:** 10.3390/ijms20235953

**Published:** 2019-11-26

**Authors:** Xue Wang, Junjie Li, Xianfeng Guo, Yan Ma, Qian Qiao, Jing Guo

**Affiliations:** 1College of Forestry, Shandong Agricultural University, No. 61, Daizong Road, Tai′ an 271018, China; 15531041099@163.com (X.W.); suduanhong@163.com (J.L.); jingguo@sdau.edu.cn (J.G.); 2Shandong Provincial Research Center of Demonstration Engineering Technology for Urban and Rural Landscape, Tai′ an 271018, China; 3Characteristic fruit tree research office, Shandong Institute of Pomology, Tai′an 271000, China; qiaoq404@163.com

**Keywords:** *PlWRKY13*, peony, *Alternaria tenuissima*, VIGS, endogenous hormones

## Abstract

Many members of the WRKY family regulate plant growth and development. Recent studies have shown that members of the WRKY family, specifically WRKY13, play various roles in the regulation of plant stress resistance. To study the function of WRKY family members in peony, the *PlWRKY13* gene (KY271095) was cloned from peony leaves. Sequence analysis and subcellular localization results revealed that *PlWRKY13* has no introns, belongs to the type IIc subgroup of the WRKY family, and functions in the nucleus. The expression pattern of *PlWRKY13* was analysed via real-time quantitative RT-PCR (qRT-PCR), which showed that the expression of *PlWRKY13* was induced by four types of abiotic stress, low-temperature, high-temperature, waterlogging and salt stress, and was positively upregulated in response to these stresses. In addition, the expression of *PlWRKY13* tended to first decrease and then increase after infection with *Alternaria tenuissima*. Virus-induced gene silencing (VIGS) technology was used to explore the function of *PlWRKY13* in the resistance of *Paeonia lactiflora* to fungal infection further, and the results showed that *PlWRKY13*-silenced plants displayed increased sensitivity to *A. tenuissima.* The infection was more severe and the disease index (DI) significantly greater in the *PlWRKY13*-silenced plants than in the control plants, and the expression of pathogenesis-related (PR) genes was also significantly altered in the *PlWRKY13*-silenced plants compared with the control plants. The contents of the endogenous hormones jasmonic acid (JA) and salicylic acid (SA) were measured, and the results showed that the JA content increased gradually after infection with *A. tenuissima* and that JA may play an active role in the resistance of *P. lactiflora* to pathogen infection, while the SA content decreased after *PlWRKY13* silencing. The contents of the two hormones decreased overall, suggesting that they are related to the transcription of *PlWRKY13* and that *PlWRKY13* may be involved in the disease-resistance pathway mediated by JA and SA. In summary, the results of our study showed that *PlWRKY13* expression was induced by stress and had a positive effect on the resistance of *P. lactiflora* to fungal infection.

## 1. Introduction

Organisms evolve to adapt to their environment, which is also the case for plants despite being sessile. Plants have evolved defence mechanisms that respond to environmental stress. These mechanisms strongly depend on the correct perception and transduction of signals through signalling cascades; thus, plant defence responses are regulated by signalling networks [1]. The transcriptional regulation of defence-related genes that respond to stress plays an important role in the development of plant stress tolerance. It has been proven that the transcriptional activation of these genes depends on the temporal and spatial function of transcription factors (TFs) [2]. Several specific DNA-binding TF families that regulate the expression of plant defence-related genes have been identified, including ERF, NAC, MYB, bZIP and WRKY proteins that bind to W-box elements [3]. WRKYs are of particular interest because they are involved in a variety of biotic and abiotic stress responses in plants and in a variety of growth and development processes. Moreover, WRKYs are highly specific and exert their regulatory activity strictly by binding to downstream promoter elements; thus, these TFs are expected to be candidate genes for crop improvement [4].

The WRKY family of TFs is one of the largest families whose members are involved in transcriptional regulation in higher plants [5], with at least 72 members in arabidopsis and more than 100 members in rice, soybean and poplar [6]. WRKYs are widely involved in a number of biological processes. For example, in arabidopsis, WRKY13 has been reported to bind to the *AtNST2* promoter and regulate the *AtNST2* gene to participate in secondary cell wall synthesis related to the development of sclerenchyma cells [7]. *AtWRKY13* and *AtWRKY12* have antagonistic effects on the regulation of the flowering period under short-day conditions: *AtWRKY13* promotes flowering, while *AtWRKY12* delays flowering [8]. There are also many studies on plant defences against biotic and abiotic stresses; the wheat transcription factors TaWRKY2, TaWRKY19, TaWRKY44, and TaWRKY93 increase salt tolerance and drought resistance by accumulating osmosis protectors (proline and soluble sugar) [9,10,11]. The ectopic overexpression of *HaWRKY76* from sunflower in *Arabidopsis thaliana* enhanced the resistance of transgenic plants to drought and waterlogging stress [12]. Studies have shown that WRKY transcription factors can also mediate a variety of disease-resistance response pathways, such as the jasmonic acid (JA) and salicylic acid (SA) signalling pathways, to regulate the expression of disease-resistance genes [13]. In resistant rice varieties, the *OsWRKY13* gene was positively regulated by the pathogens *Rhizoctonia solani* and *Sarocladium oryzae* [14]. The overexpression of a pair of alleles, *OsWRKY45-1* and *OsWRKY45-2*, showed that these two alleles had a positive regulatory effect on the resistance of rice to the fungal pathogen *Magnaporthe grisea* [13]. The expression of the pathogenesis-related (PR) genes, including *PR1*, *PR2*, *PR4* and *PR5*, in transgenic tobacco ectopically overexpressing cotton *GhWRKY15* was higher than that in the control and enhanced the disease resistance of the transgenic plants; the response mechanism of *GhWRKY15*-overexpressing plants to viral and fungal infections may be related to JA, SA or ethylene (ET) signalling pathways [15]. In addition, *AtWRKY70* mediates the antagonistic effect between SA and JA and integrates signals from the two pathways as an activator of SA-inducing genes and an inhibitor of JA-responsive genes [16]. 

WRKY proteins are named for their highly conserved amino acid WRKYGQK heptapeptide sequence, which contain zinc-finger-like motifs, Cys(2)-His(2) or Cys(2)-HisCys, that can bind to the DNA sequence W-box (TTTGACC/T) [17]. WRKY transcription factors can be divided into three groups according to the number of DNA-binding domains and the characteristics of the zinc-finger-like motifs. Group I is characterized by two different domains: N-terminal and C-terminal motifs; group II is the largest, with only one WRKY motif, and the zinc-finger-like motif of group II is the same as that of group I, Cys(2)-His(2). Group II was initially divided into five subgroups, IIa, IIb, IIc, IId and IIe, but recent phylogenetic analysis showed that IIa and IIb, IIc and IId combined into IIa+b and IIc+d, respectively [18]. Group III has different zinc-finger-like motifs, namely, the Cys(2)-HisCys motif [19].

As an important ornamental flower worldwide, peony (*Paeonia lactiflora* Pall) is often restricted by different environmental stresses, such as temperature, high salt and waterlogging. In production, peony is also prone to the red spot disease caused by *Alternaria tenuissima*, which severely affects the ornamental value and economic value of peony [20]. Previous reports have shown that *WRKY* genes play an important role in resistance to abiotic and biotic stresses. In this study, on the basis of transcriptome data from *P. lactiflora*, we cloned the *PlWRKY13* gene, which is related to various stresses, and discussed its role in defence mechanisms via evolutionary and functional analyses, which is of great significance for the study of the *P. lactiflora* WRKY family. Our study also expands the field of functional research of WRKY family members.

## 2. Results and Analysis

### 2.1. Cloning and Sequence Analysis of PlWRKY13

*P. lactiflora* cDNA was used as a template to design specific primers for open reading frame (ORF) amplification. The results showed that the *PlWRKY13* ORF was 693 bp long and encoded 230 amino acids. Then, total DNA of *P. lactiflora* was used as a template for PCR amplification to obtain the DNA sequence of *PlWRKY13* from the start codon to the stop codon. DNAMAN 5.0 software was used to compare DNA sequences with cDNA sequences, and it was determined that the sequence alignment results for *PlWRKY13* were consistent, that is, there were no introns (Figure 1A). According to the classification standard of *Arabidopsis* WRKYs by Eulgem et al. [21], the WRKY family type to which the PlWRKY13 belongs was determined, and the phylogenetic tree of PlWRKY13 and *Arabidopsis* WRKY family genes was constructed for phylogenetic analysis and clustering (Figure 1B). The results showed that PlWRKY13 and AtWRKY13 were a branch of the type IIc subgroup of the WRKY family. 

Multiple comparative analyses showed that the coding region of the *PlWRKY13* gene contained a WRKY domain composed of 58 amino acids (position 152-209) and had high homology with the conserved domain of WRKY13 transcription factors cloned from other plants. The N-terminus of its conserved domain contains the highly conserved WRKYGQK heptapeptide sequence, and the C-terminus has a C_2_H_2_ (CX_5_CX_23_HNH) zinc finger structure (Figure 1C). To understand the relationship further between the PlWRKY13 protein and the WRKY13 proteins of other plant species, WRKY13 amino acid sequences from 10 plant species that were highly homologous (>50%) to PlWRKY13 were selected and constructed into an evolutionary tree. The results showed that, among the 11 WRKY13 proteins analysed (Figure 1D), *P. lactiflora* PlWRKY13 was most closely related to *Morus notabilis* MnWRKY13, and was classified into one branch. Subsequently, this branch was closely related to *Sesamum indicum* SiWRKY13, and both were classified into a branch. *Jatropha curcas* JcWRKY13, *Citrus sinensis* CsWRKY13, *Populus euphratica* PeWRKY13 and *Ricinus communis* RcWRKY13 were classified as one branch, and *Theobroma cacao* TcWRKY13, *Gossypium hirsutum* GhWRKY13, *Cajanus cajan* CcWRKY13 and *Glycine soja* GsWRKY13 were classified as one branch; these two branches were classified as one branch, which was distantly related to *P. lactiflora*.

### 2.2. Subcellular Localization

To determine the localization of the PlWRKY13 protein in cells, the constructed pROKII-*PlWRKY13*-GFP fusion expression vector was used as the test material, and the empty vector containing only GFP was used as the control. The two vectors were transferred into the epidermal cells of tobacco leaves, and the results showed that the green fluorescence in the epidermal cells of tobacco leaves transformed with the recombinant vector was present only in the nucleus, while the fluorescence in the leaves of the control group was widely distributed, with fluorescence in the nucleus and cytoplasm (Figure 2). Therefore, the PlWRKY13 protein is located in the nucleus.

### 2.3. Expression Patterns of PlWRKY13 under Different Stresses

Under four abiotic stresses, low-temperature, high-temperature, waterlogging and salt stress, *PlWRKY13* was positively regulated in response to the stress conditions (Figure 3A–D). Among them, the response of *PlWRKY13* was the fastest when induced by low temperature, and the expression of *PlWRKY13* was extremely significantly higher (*p <* 0.01) under low-temperature conditions than under the control conditions 2 h after stress treatments. The response rate under high-temperature and waterlogging stress conditions was intermediate, and the expression of *PlWRKY13* was extremely significantly higher (*p <* 0.01) under high-temperature and waterlogging stress conditions than that under the control conditions 4 h after treatment. However, *PlWRKY13* showed a relatively slow response under salt stress and had extremely significantly higher expression (*p <* 0.01) under salt-stress conditions than under the control conditions 8 h after stress, *PlWRKY13* expression reached a peak at 12 h, and then returned to the normal level. In response to infection with *A. tenuissima*, *PlWRKY13* tended to decrease and then increase in expression (Figure 3E). The expression of the *PlWRKY13* gene in the leaves infected with *A. tenuissima* decreased rapidly in the first 12 h and then gradually increased and recovered to the same level as that in the control. Then, at 96 h after infection, the *PlWRKY13* expression level reached a peak that was 1.65 times higher than that of the control (*p <* 0.01). These results suggest that *PlWRKY13*, as a transcriptional activator, regulates the response of *P. lactiflora* to high-temperature, low-temperature, waterlogging and salt stress and infection with *A. tenuissima* by improving its transcriptional level.

### 2.4. VIGS Treatment Reduced the Transcription of the Endogenous PlWRKY13 Gene

We selected the fragment composed of the first 348 bases in *PlWRKY13* gene as the target sequence, because it does not have homology of the WRKY family. cDNAs from the leaves treated with the empty vector TRV::00 and TRV::WRKY13 for 10 days were used as templates, and specific primers were designed for the on vectors pTRV1 and pTRV2 (Figure 4A). PCR was used to detect whether tobacco rattle virus (TRV) was inserted into the genome of *P. lactiflora* and expressed. The results showed that 647-bp and 372-bp target bands were amplified from plants treated with the empty vector, while 647-bp and 720-bp target bands were amplified from plants treated with TRV::WRKY13 (Figure 4B). This indicated that TRV::00 and TRV::WRKY13 had been successfully inserted into the genome of *P. lactiflora* and expressed. When plants were silenced for 16 days, the cDNAs from the plants treated with the blank control, the empty vector and TRV::WRKY13 were used as templates. The quantitative real-time PCR (qRT-PCR) results showed that the transcription abundance of the *PlWRKY13* gene in TRV::WRKY13-treated plants was significantly reduced by 40%-60% (*p <* 0.01) compared with that in the blank control plants, while the transcription level of the *PlWRKY13* gene in the empty vector-treated plants was not significantly different (*p >* 0.05) from that of the blank control plants (Figure 4C). This suggests that *PlWRKY13* was effectively silenced in plants treated with TRV::WRKY13.

### 2.5. PlWRKY13-Silenced Plants Exhibited Increased Sensitivity to A. tenuissima

Plants silenced for 20 days (silencing efficiency > 60%) and the blank control plants were selected for infection with *A. tenuissima*. The results showed that the expression level of *PlWRKY13* in the gene-silenced plants was extremely significantly lower (*p <* 0.01) than that in the control group, but surprisingly, the expression trend of *PlWRKY13* in the two groups was consistent (Figure 5A). This indicated that the TRV-PlWRKY13-silencing vector effectively inhibited the expression of *PlWRKY13* in *P. lactiflora*, and in silenced plants, *PlWRKY13* was still responsive to *A. tenuissima* infection.

After infection with *A. tenuissima*, we found that *PlWRKY13*-silenced plants were more sensitive to the pathogenic fungus (Figure 5B) and exhibited more serious disease conditions than the blank control and the empty vector control. *PlWRKY13*-silenced plants had large disease spots areas, and the infected parts were scorched and brittle, with the appearance of perforation in serious cases. The disease incidence statistics were carried out 21 days after the plants were infected with *A. tenuissima*. The results showed that all plants were sick, and the disease index (DI) of the blank control, empty vector control and *PlWRKY13*-silenced plants were 43.05, 44.10 and 55.14, respectively (Figure 5C). The disease index of the silenced plants was extremely significantly higher (*p <* 0.01) than that of the other two groups, indicating that *PWRKY13* is involved in the resistance of *P. lactiflora* to *A. tenuissima*. There was no significant difference (*p >* 0.05) in the phenotypic characteristics and disease index between the blank control group and the empty vector control group, indicating that the TRV vector itself had no significant impact on peony. 

### 2.6. PlWRKY13 Silencing Decreased the Endogenous JA and SA Concentrations

To investigate the response of endogenous hormones in the leaves of *P. lactiflora* to pathogen infection and the relationship between *PlWRKY13* and endogenous hormones, the contents of endogenous JA and SA in the leaves at 0, 12, 24, 48, 72 and 96 h after infection with *A. tenuissima* were detected in this study. The results showed that the JA content generally increased from 0 h to 96 h after infection, reaching a peak at 96 h, while the SA content decreased from 0 h to 48 h, then increased, peaked at 72 h, and decreased to below the initial level. The change in JA and SA in response to pathogen infection in *PlWRKY13*-silenced plants was basically the same as that in the control plants, except that the overall content of these two hormones in the silenced plants was lower than that in the control plants (Figure 6A). These results showed that JA and SA in *P. lactiflora* were closely related to the disease-resistance process and were correlated with the transcription level of *PlWRKY13.* The silencing of *PlWRKY13* may inhibit the disease-resistance regulation pathway mediated by JA and SA.

### 2.7. Differential Regulation of PR Genes by PlWRKY13

The silencing of the *PlWRKY13* gene resulted in decreased levels of JA and SA; therefore, the resistance of *PlWRKY13*-silenced plants to *A. tenuissima* decreased, which may also be involved in the regulation of genes in the JA and SA signalling pathways. To test this hypothesis, the expression of PR genes in the JA and SA signalling pathways was analysed by qRT-PCR. The expression levels of PR1, PR2, PR4B, PR5 and PR10 were detected and analysed after *A. tenuissima* infection in *PlWRKY13*-silenced plants and control plants for 96 h. The results showed that the expression levels of PR2, PR4B, PR5 and PR10 in the *PlWRKY13*-silenced plants were extremely significantly lower (*p <* 0.01) than those in the blank control and empty vector control, and the expression level of PR1 in the *PlWRKY13*-silenced plants was significantly higher (0.01 < *p <* 0.05) than that in the blank control and empty vector control; there was no significant difference (*p >* 0.05) between the blank control and empty vector control (Figure 6B). Differences in the expression of PR genes suggest that their expression is affected by *PlWRKY13* silencing, which further suggests that *PlWRKY13* may enhance the resistance of *P. lactiflora* to *A. tenuissima* by participating in the JA- and SA-mediated disease-resistance signalling pathways.

## 3. Discussion

Although some transcription factor families in plants have been shown to play important roles in regulating plant defence responses, such as bZIP, MYB, WRKY, and EREBF, the exact functions and mechanisms of individual transcription factors are poorly understood [22]. Many studies have found that the functions of WRKYs are related to the growth and development of plants and defence against external stresses, but these studies mostly focus on Arabidopsis, rice, cotton and other crops [15,23,24], while few studies have focused on members of the peony WRKY family. Therefore, we isolated the PlWRKY13 transcription factor from the important ornamental horticultural plant *P. lactiflora* and studied its biological characteristics and functions.

The isolated *WRKY* gene was compared with members of the *A. thaliana WRKY* family, and both the *PlWRKY13* gene and *AtWRKY13* belonged to the type IIc subgroup of the WRKY family. Protein sequence analysis revealed that PlWRKY13 has a conserved domain highly homologous to WRKY13 transcription factors in other plants. The N-terminus of the conserved domain has a WRKYGQK heptapeptide sequence, and the C-terminus has C_2_H_2_ zinc finger structures typical of Group I and Group II WRKY transcription factors [18]; PlWRKY13 is most closely related to *M. notabilis* MnWRKY13. The orderly distribution and dynamic regulation of proteins are the basis of ensuring the growth and development of living organisms [25], therefore, subcellular localization is an essential step for functional analysis. The results showed that, similar to most WRKY proteins in previous studies [15,26,27], PlWRKY13 is located to the nucleus and has the general characteristics of TFs. Therefore, we hypothesized that, similar to most WRKY proteins, the PlWRKY13 protein activates the transcription of its target gene by binding to the W-box element of its promoter.

Many members of the WRKY family are involved in the regulation of plant biotic and abiotic stresses. For example, under osmotic stress and salt stress, *AtWRKY46* promotes the growth of lateral roots through abscisic acid (ABA) signalling and auxin homeostasis, thereby regulating resistance to stress conditions [28]. *BhWRKY1* can bind with the *BhGolS1* promoter to participate in the regulation of drought resistance and cold tolerance of *Boea hygrometrica* [29]. The expression of multiple *PsWRKYs* in *Papaver somniferum* can be induced under trauma, cold, drought, salt and MeJA stresses [30]. In this study, the expression of *PlWRKY13* was upregulated under four abiotic stresses, but its responsiveness and degree of response were different under different stress treatments. Among them, the response to low temperature was the fastest, and *PlWRKY13* expression showed significant differences 2 h after treatment; salt stress induced the slowest response, showing a difference in expression at 8 h after treatment. Interestingly, *PlWRKY13* expression was most sensitive to salt stress induction, reaching a peak at 12 h, at which time *PlWRKY13* expression under salt stress was 9.07 times higher that under the control conditions; however, under high-temperature and waterlogging stress, *PlWRKY13* presented significant differences 4 h after treatment. Moreover, the expression of *PlWRKY13* exhibited an upward-downward-upward trend under high temperature, which may be related to changes in endogenous hormone contents. High temperature seems to induce an increase in JA and a decrease in ABA, moreover, JA may inhibit the expression of *PlWRKY13*, leading to the down-regulation of gene expression, which needs to be studied further. These results indicate that *PlWRKY13* is involved in the response of peony to abiotic stress. However, the overexpression of rice *OsWRKY13* was significantly inhibited after 12 h-5 days of drought stress and 3–36 h of high-salt stress [31]. *OsWRKY13* seems to preferentially bind to the promoter of downregulated genes in vitro, which indicates that it may play a negative transcriptional regulatory role [32], although its positive transcriptional regulation function cannot be ruled out.

The involvement of WRKY proteins in plant disease resistance has been widely reported in Arabidopsis, cotton, rice and other plants. *AtWRKY52* contains a Toll/interleukin-1 receptor–nucleotide-binding site-leucine-rich repeat (TIR-NBS-LRR) domain, which can interact with RPS4 to regulate resistance to the fungal pathogen *Colletotrichum higginsianum* and the bacterial pathogen *Pseudomonas syringae* [33]. After treatment with SA, MeJA and ET, the expression of the *GhWRKY40* gene was upregulated and regulated the resistance of cotton to *Ralstonia solanacearum* [34]. However, in tomato, *SlWRKY70* transcription exhibited negative regulation of plant resistance to fungal infection [35]. Similarly, in island cotton, *GbWRKY1* mediated cotton resistance through the JA signalling pathway and negatively regulated cotton resistance to *Botrytis cinerea* and *Verticillium dahliae* [36]. Therefore, WRKY transcription factors can be positively or negatively regulated to participate in plant disease resistance. *Erwinia carotovora* infection in wild-type *Arabidopsis* can induce increased *AtWRKY70* gene expression, and this increase has been proven to be associated with elevated levels of endogenous SA. At the initial stage of infection, the transient increase in JA inhibits the expression of *AtWRKY70*. Studies have concluded that *AtWRKY70* is a transcriptional activator of SA-inducing genes and a transcriptional repressor of JA-inducing genes, leading to the intersection of the SA- and JA-mediated signal defence pathways [16]. In this experiment, the response of *PlWRKY13* to *A. tenuissima* showed negative expression first and then positive expression regulation, which was 1.65 times higher than that of the control group at 96 h and possibly the same as that of *Arabidopsis*, due to the inhibition by JA at the early stage of expression. To explore the function of *PlWRKY13* in disease resistance, we performed additional experiments.

The interaction between plants and pathogens is a very complex process, involving many small molecule signal substances and electrical signals, which jointly participate in the formation of plant disease resistance network. Plant hormones are key factors in regulating the response of plant cells to external and internal stimuli, and they can interact with each other to regulate the response of cells to external changes [37]. Among them, JA- and SA- induced plant disease resistance are widely distributed. When plants are attacked by pathogens, the contents of endogenous JA and SA will be rapidly increased by disease-resistance regulatory pathways and then will induce the transcriptional regulation of multiple disease-resistance genes [38]. In the complex network regulated by plants in response to external stress, there is a significant antagonistic relationship between SA and JA signalling pathways, studies have shown that SA and JA may inhibit each other′s synthesis and their downstream signalling pathways [37], moreover, node genes in SA and JA signalling pathways, such as *NPR1* and *SSI1*, may play fine regulatory roles in their antagonistic effects [39], which confirms the completely different changes in SA and JA after infection with *A. tenuissima* in this experiment. However, studies have also found that JA and SA have a synergistic regulatory effect, but the specific regulatory mechanism is still unknown [40]. The external application of SA and MeJA in bananas significantly reduced the disease index of anthracnose disease, indicating that SA and MeJA induced the resistance of banana fruits to *Colletotrichum musae* [41]; the same conclusion was drawn successively from apples, citrus and peaches [42,43,44]. Some WRKYs and PR genes are highly sensitive to response hormones. For example, *MaWRKY1*, *MaWRKY2*, *MaPR1-1*, *MaPR2* and *MaPR10c* are all induced and enhanced in banana with the accumulation of endogenous SA and JA contents, *MaWRKY1* and *MaWRKY2* can bind to the promoters of the *MaPR1-1*, *MaPR2* and *MaPR10c* genes to regulate disease resistance, while *MaPR5-2* and *MaPR5-3* are induced only by MeJA [41]. Previous research has shown that *MaPR1-2* and *MaPR1-3* are induced by SA and MeJA to reduce transcription abundance, which reflects the results of *PlPR1* in this study [44,45]. However, pathogens, SA and JA can induce the upregulation of PR genes in most cases, and these downregulated PR genes need to be further studied.

Unlike *AtWRKY70*, which is activated by SA and inhibited by JA [16], rice *OsWRKY13* is co-induced by SA and JA, and surprisingly, studies have showed that *OsWRKY13* affected SA accumulation. The *OsWRKY13*-overexpressing plants accumulated more free SA than the wild-type plants, while the wild-type plants had more SA β-glucoside than the *OsWRKY13*-overexpressing plants. Moreover, *OsWRKY13* affects the expression of defence-related genes in SA- and JA-dependent signalling, including the *OsPR1a* and *OsPR10* genes. The overexpression of *OsWRKY13* can induce a significant increase in *OsPR1a* but inhibit the expression of *OsPR10*. Therefore, it is believed that one of the action sites of *OsWRKY13* may be located downstream of SA and JA and upstream of the PR protein in the defence signal network [46]. In this study, the silencing of *PlWRKY13* led to a decrease in endogenous JA and SA levels, confirming the results in rice. Considering previous studies, we speculated that *PlWRKY13* may be involved in the SA- and JA-mediated defence signalling pathways, and has direct or indirect feedback regulation on the content of these two hormones. Moreover, silencing *PlWRKY13* resulted in changes in the expression of 5 PR genes. In addition to *PlPR1*, the expression of *PlPR2*, *PlPR4B*, *PlPR5* and *PlPR10* was downregulated, indicating that *PlWRKY13* may directly or indirectly induce the expression of *PlPR2*, *PlPR4B*, *PlPR5* and *PlPR10* through SA or JA signalling pathways and inhibit the expression of *PlPR1*.

There are few studies on the functions of the WRKY family in peony. In this experiment, we have tried various methods for silencing peony *PlWRKY13*, and finally adopted the vacuum suction filtration under negative pressure method with the highest silencing efficiency. As a convenient technique to explore gene function widely used in recent years, VIGS is more intuitive in phenotypic variation, especially applicable to species with unstable genetic transformation system, such as grapevine [47], citrus [48], tomato [49], gladiolus [50], etc. However, the use of VIGS still has some disadvantages, including the limited duration of silence, different viral vectors will produce different silence efficiency for different species [48]. In our study, we used VIGS technology to preliminarily explore the function and regulatory role of *PlWRKY13* in response to *A. tenuissima*, and its specific mechanism still needs to be further explored.

In this study, we analysed the functions of the *PlWRKY13* gene in peony under stress conditions and found that it positively regulated the resistance to low-temperature, high-temperature, waterlogging and salt stress. Moreover, *PlWRKY13* may be involved in the SA- and JA-mediated signal defence pathways to resist disease infection when *P. lactiflora* is infected with *A. tenuissima*, and it may be located upstream of PR genes, regulating the expression pattern of specific PR genes to further regulate the disease-resistance mechanism of peony.

## 4. Materials and Methods

### 4.1. Plant and Fungal Materials

*P. lactiflora* ′Da Fugui′ was planted in pots outside the forestry experimental station of Shandong Agricultural University, Tai’ an, Shandong, China. We planted the peony buds in flower pots in autumn and selected robust and consistent plants for the experiments in the following spring. Leaves were selected as experimental materials, and the collected leaves were quickly frozen in liquid nitrogen and stored at −80 °C. Three biological replicates were included.

*Nicotiana benthamiana* was cultured on substrate and placed in a light incubator (25 °C, 120 μmol·m^−2^·s^−1^, with a light dark period of 16 h/8 h) for constant temperature culture. When the plant grew to 8 true leaves, the subcellular localization test was carried out.

*A. tenuissima* was isolated from infected peony leaves and cultured in potato dextrose agar (PDA) solid medium for 7 days at 28 °C. The colony margins (cake 6 mm in diameter) on PDA solid medium were then cut out as the source of infection.

### 4.2. Total RNA Extraction and cDNA Synthesis

Total RNA of all plant materials was extracted by referring to the instructions of the Aidlab EASY spin rapid RNA extraction kit (Aidlab Biotech, Beijing, China). The synthesis of the first strand of cDNA was performed with the ComWin Biotech reverse transcription kit (ComWin Biotech, Beijing, China).

### 4.3. Cloning and Sequence Analysis of PlWRKY13

According to the unigene functional annotation of the WRKY gene in the transcriptome data of *P. lactiflora* ′Da Fugui′, the *PlWRKY13* gene was screened out to predict its ORF sequence, and specific primers were designed (Table 1). Total leaf DNA was extracted via the improved CTAB method, and the full-length *PlWRKY13* gene was cloned using it as a template. WRKY13 proteins of several different plant species having high amino acid sequence homology (> 50%) with peony PlWRKY13 were selected from GenBank, after which multiple alignment was performed by DNAMAN5.0 software to analyse the structures of the gene domains and conserved domains. Phylogenetic analysis was then performed via MEGA 5.0 software, and neighbour-joining evolutionary tree was subsequently constructed. Bootstraps were used to test the evolutionary tree, with a total of 1000 repetitions.

### 4.4. Subcellular Localization

Full-length cDNA with the termination codon removed from *PlWRKY13* was used as a template, and specific primers with restriction sites (XbaI and KpnI) were designed for PCR amplification (Table 1). After enzyme digestion, the obtained product was ligated to the pROKII-GFP vector that was also double digested, and the fusion expression vector pROKII-*PlWRKY13*-GFP was verified by sequencing. The recombinant vector (pROKII-*PlWRKY13*-GFP) and control vector (pROKII-GFP) were introduced into tobacco leaves through Agrobacterium infection. After culture for 3 days, the GFP fluorescence of samples was observed under a Nikon C2-ER confocal laser scanning microscope (Nikon, Tokyo, Japan).

### 4.5. Stress Treatments and Infection with A. tenuissima

Peonies at the middle stage of leaf development with consistent growth were placed in the culture room with a 4 °C and 40 °C climate to simulate low- and high-temperature conditions (the relative humidity was 70%, the light dark period was 16 h/8 h, and the light intensity was 180 umol m^−2·^s^−1^). Some peonies were immersed in a container full of water to simulate waterlogging pressure, some peonies were treated with a 200 mmol·L^−1^ NaCl solution to simulate salt stress, and other peonies with consistent growth served as blank controls, that is, they did not receive any treatment (the temperature for waterlogging and salt stress and blank control was 25 °C). Leaves at 0, 2, 4, 8, 12 and 24 h after stress treatment and those of the blank controls were collected as test materials, and the growth sites of leaves were ensured to be consistent during sampling. A sterilized needle was used to make micro-wound on the surface of the leaves, and an *A. tenuissima* infection source was placed on the micro-wound for infection treatment, while the same size of the water agar block was placed on the micro-wound for blank control treatment [51]. After 0, 12, 24, 36, 48, 72 and 96 h, the leaves of the experimental group and of the control group were collected respectively. The incidence grade of peony was calculated according to Li [20], and the corresponding disease index was as follows: Disease Index (DI) = ∑ (number of disease-grade plants × the representative value) × 100% / plant number × the representative value of the most serious disease. All of the above samples were frozen in liquid nitrogen and stored at −80 °C. Three biological replicates included per treatment.

### 4.6. VIGS in P. lactiflora

To enable the *PlWRKY13* gene to be specifically silenced, a 348-bp non-conserved fragment of the *PlWRKY13* gene was selected and in-frame cloned into the pTRV2 vector. The correct recombinant vector was verified by PCR and sequencing.

The pTRV1, pTRV2 and pTRV2-WRKY13 plasmids were transformed into *A. tumefaciens* GV3101 competent cells by freeze-thaw method. A total of 1 mL of *A. tumefaciens* GV3101-pTRV1, GV3101-pTRV2 and GV3101-pTRV2-WRKY13 was cultured in 10 mL yeast extract peptone sucrose (YEP) liquid medium (including Kan 50 μg/mL and Rif 100 μg/mL) at 28 °C for 24 h at 200 r·min^−1^. After that, 10 mL of bacterial liquid was transferred to 400 mL YEP liquid medium (including Kan 50 μg/mL, Rif 100 μg/mL and acetosyringone (AS) 200 μM), and cultured at 28 °C for 5–6 days at 200 r·min^−1^. When the OD600 of the bacterial liquid was approximately 1.5, the bacteria were centrifuged at 4 °C and 12000 r·min^−1^ for 2 min, collected, and suspended in solution (MES 10 mmol/L, MgCl_2_ 10 mmol/L, AS 150 uM and aseptic water as the solvent), and the concentration of the suspension was adjusted to approximately 1.5. GV3101-pTRV2 and GV3101-pTRV2-WRKY13 were mixed with GV3101-pTRV1 in a 1:1 ratio and placed at rest for 3–5 h at room temperature in darkness [52].

The plants with strong growth potential and a consistent growth state were selected, and VIGS infection was carried out by vacuum suction filtration under negative pressure. The specific method is to soak the plant in the infection solution in a vacuum bucket when the underground buds are just beginning to differentiate, and then infecting the plant. The treated plants were replanted in pots and left in the dark for 24 h, and normal field management of the treated and control plants was carried out.

### 4.7. Quantitative Real-Time PCR (qRT-PCR)

In this experiment, qRT-PCR was used to determine gene expression. The instrument was a Bio-Rad CFX96™ Real-Time system (Bio-Rad, Hercules, CA, USA), and the qRT-PCR mixture (total volume of 20 μL) contained 10 μL of SYBR^®^ Premix Ex Taq™ (TaKaRa, Kyoto, Japan), 8 μL of ddH_2_O, 0.5 μL of each primer and 1 μL of cDNA. The reaction procedure was as follows: 95 °C for 30 s; 40 cycles of 95 °C for 5 s and 60 °C for 30 s; and then a dissociation stage of 95 °C for 10 s, 65 °C for 5 s and 95 °C for 5 s. *PlActin* was used as the internal controls gene, each sample set included three biological replicates, and data analysis was performed using the 2^–ΔΔCT^ method [53].

### 4.8. Determination of Endogenous Hormones

Sampling was performed on *P. lactiflora* leaves at 0, 12, 24, 48, 72 and 96 h after pathogen infection, and the contents of the endogenous hormones JA and SA were determined by high-performance liquid chromatography [54]. The chromatographic conditions were as follows: For JA, the high-performance liquid chromatography instrument was a RIGOL L3000 (RIGOL, Suzhou, China) with a wavelength of 210 nm and a Kromasil C18 reversed-phase chromatographic column (250 mm × 4.6 mm, 5 micron); the flow rate was 0.8 mL/min; the mobile phase was 1% phosphoric acid aqueous solution:acetonitrile = 45:55 (*V*/*V*); and the sample volume was 10 μL. For SA, the high-performance liquid chromatography instrument was a Waters 1525 (Waters, Shanghai, China) with a fluorescence detector, an excitation wavelength of 294 nm, an emission wavelength of 426 nm, and a Kromasil C18 reversed-phase chromatographic column (250 mm × 4.6 mm, 5 micron); the flow rate was 0.8 mL/min; the mobile phase was 1% acetic acid solution:methanol = 2:3 (*V*/*V*); and the sample volume was 10 μL. Three biological replicates were performed for each sample.

### 4.9. Statistical Analysis

At least three biological replicates were included in the data, and all data were analysed using ANOVA and Student′s *t*-test for the determination of the significant differences with SPSS24.0 software (SPSS Inc., Chicago, IL, USA).

## Figures and Tables

**Figure 1 ijms-20-05953-f001:**
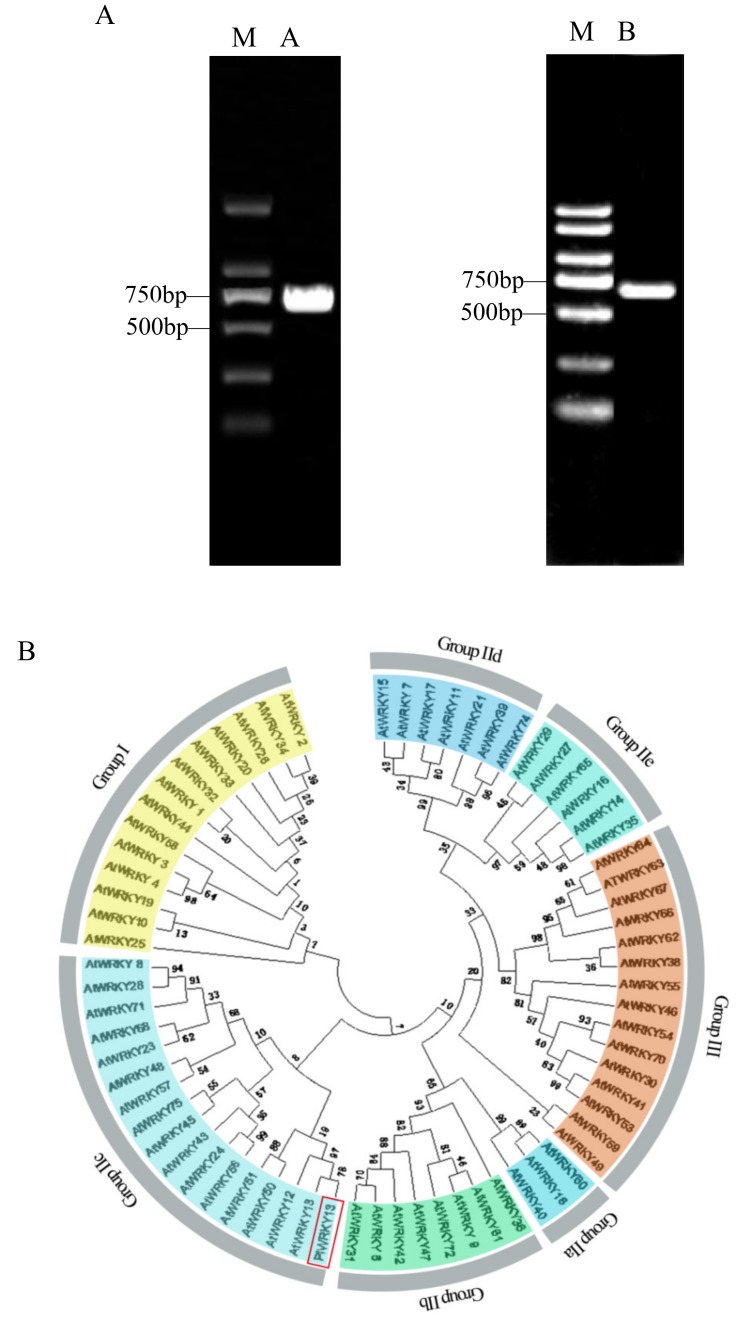
Sequence and evolutionary analysis of PlWRKY13. (**A**) PCR amplification products of *PlWRKY13*: M, DL2000 DNA marker; A, *PlWRKY13* cDNA fragment; B, *PlWRKY13* fragment. Different backgrounds distinguish different subgroups, the red box is peony PlWRKY13. (**B**) Phylogenetic analysis based on the amino acid sequences of PlWRKY13 and WRKY family proteins in *Arabidopsis*. (**C**) Multiple alignment of the deduced PlWRKY13 amino acid sequences with its homologues. Note: The box indicates the WRKYGQK heptapeptide sequence, and the triangle indicates the zinc finger structure. Different colors represent homology difference, among which dark blue parts have the highest homology, pink parts have the second highest homology, light blue parts have the lowest homology, and the red box is WRKYGQK heptopeptide sequence. (**D**) The phylogenetic tree derived from the alignment of the amino acid sequences of PlWRKY13 and other WRKY13. In the red frame is Paeonia lactiflora PlWRKY13 protein.

**Figure 2 ijms-20-05953-f002:**
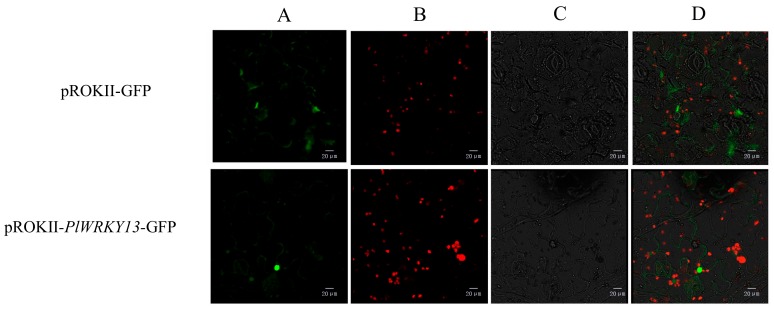
Subcellular localization of PlWRKY13 in leaves of *Nicotiana benthamiana.* (**A**) Fluorescent vision. (**B**) Chloroplast auto-fluorescent vision. (**C**) Vision under nature light. (**D**) Overlapped vision.

**Figure 3 ijms-20-05953-f003:**
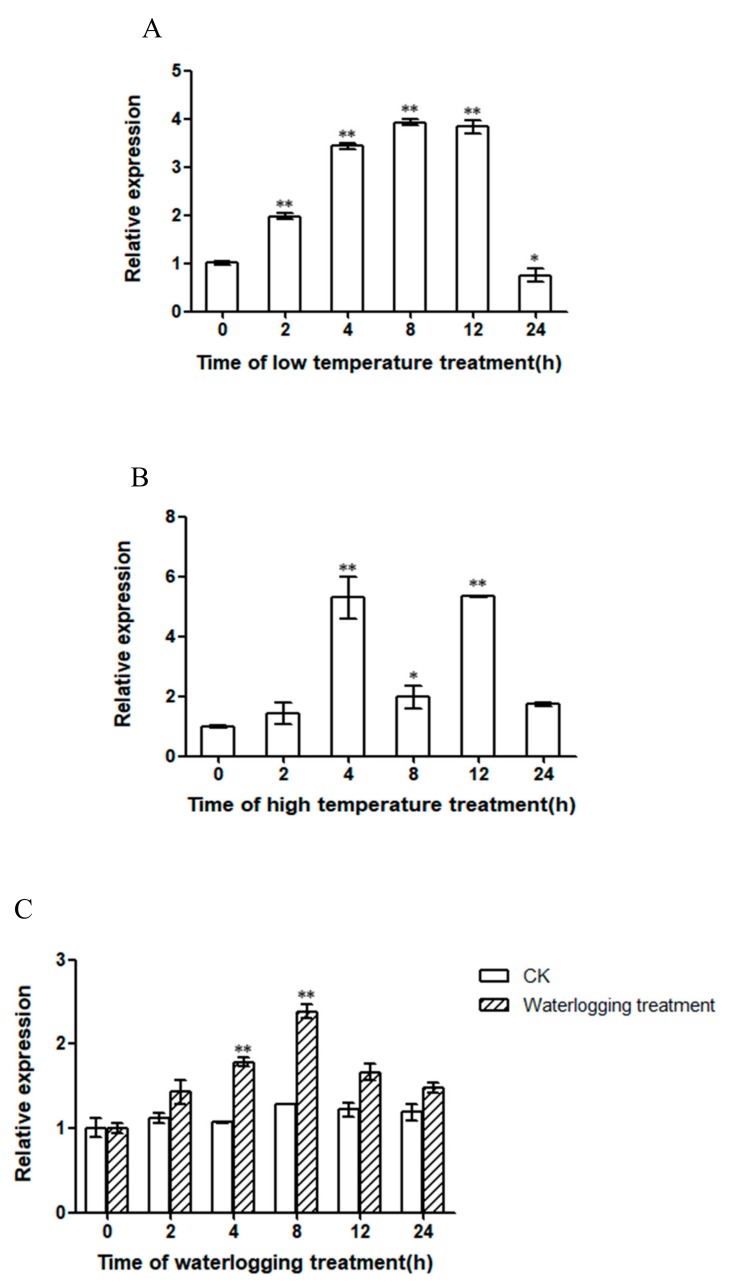
Expression patterns of *PlWRKY13*. (**A**) Low temperature stress. (**B**) High temperature stress. (**C**) Waterlogging stress. (**D**) Salt stress. (**E**) Infection with *A. tenuissima.* * indicates a significant difference between the treatment and control (0.01 < *p <* 0.05); ** means an extremely significant difference (*p <* 0.01).

**Figure 4 ijms-20-05953-f004:**
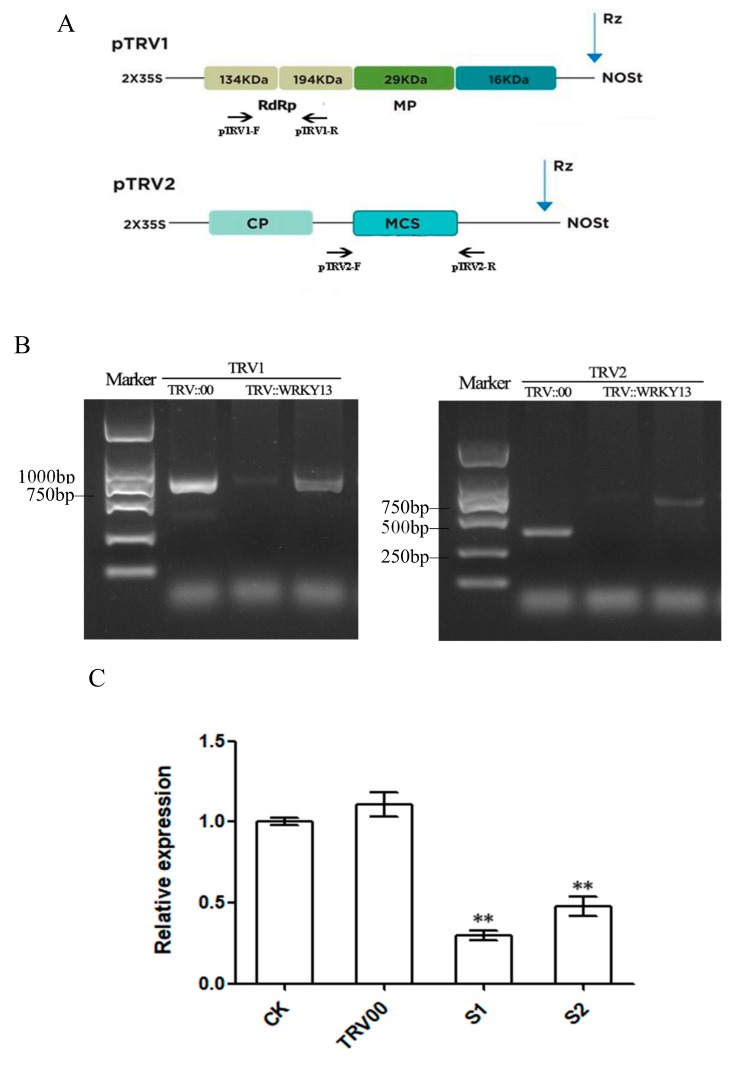
Molecular detection of tobacco rattle virus (TRV) and expression analyses of *PlWRKY13* in virus-induced gene silencing (VIGS) plants. (**A**) Plasmid pTRV1 and pTRV2 and primer schematic diagram. The pTRV1/2-F/R arrows represent the primer design sites, and the Rz arrows represent the ribozymes insertion sites. (**B**) The PCR detection of RNA1 and RNA2 of TRV in *P. lactiflora* leaves. (**C**) The gene expression level of *PlWRKY13*-silenced plants. S1 and S2 are two independent repeating TRV::WRKY13 individuals, * indicates a significant difference between the treatment and control (0.01 < *p <* 0.05); ** means an extremely significant difference (*p <* 0.01).

**Figure 5 ijms-20-05953-f005:**
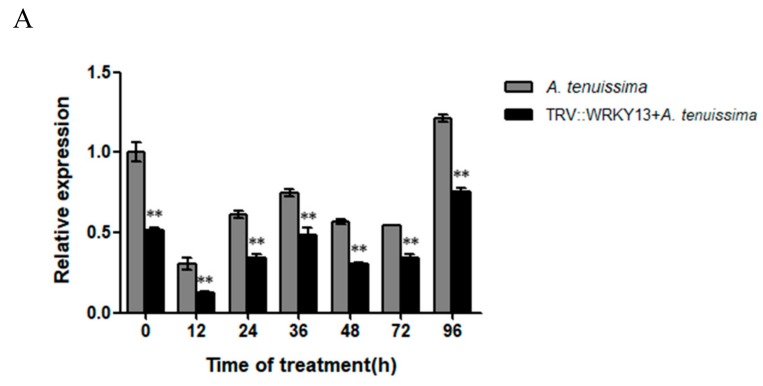
Phenotypic differences in *PlWRKY13*-silenced plants in response to *A. tenuissima.* (**A**) Expression of *PlWRKY13* in response to pathogen after silencing. (**B**) Leaf phenotype. (**C**) Disease index. * indicates a significant difference between the treatment and control (0.01 < *p <* 0.05); ** means an extremely significant difference (*p <* 0.01).

**Figure 6 ijms-20-05953-f006:**
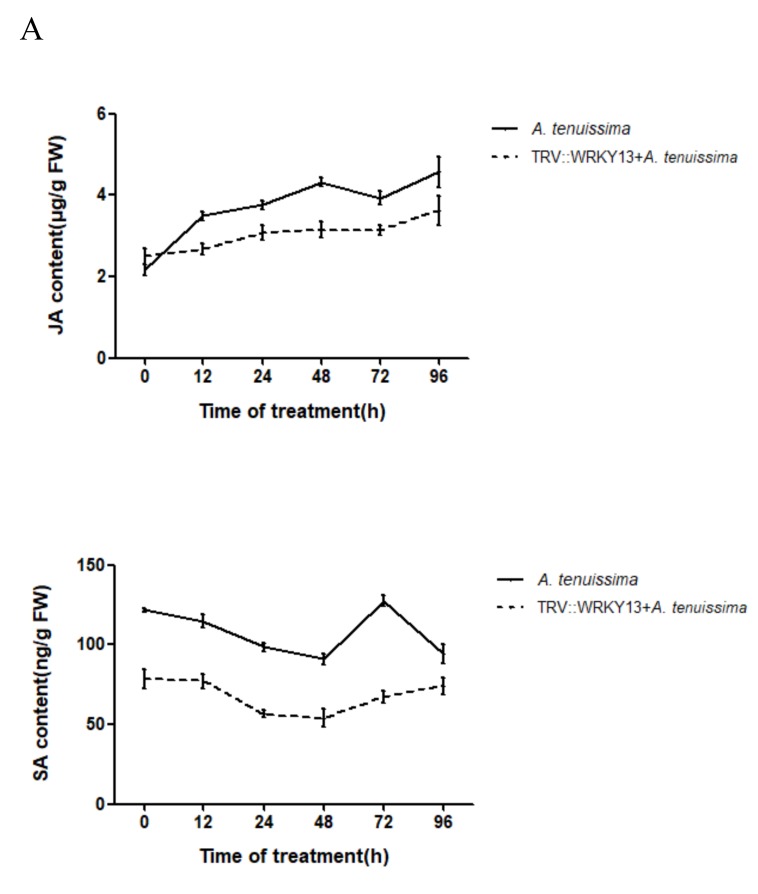
Endogenous hormone concentrations and the expression of PR genes. (**A**) Endogenous hormone contents in the leaves of *P. lactiflora* infected with *A. tenuissima*. (**B**) Expression of PR genes in the control and gene-silenced leaves. * indicates a significant difference between the treatment and control (0.01 < *p <* 0.05); ** means an extremely significant difference (*p <* 0.01).

**Table 1 ijms-20-05953-t001:** Primers and their sequences used in this study.

Primer Name	(5’ 3’) Nucleotide Sequence *	Purpose
PlActionF	ACTGCTGAACGGGAAATT	*Actin* Primers
PlActionR	ATGGCTGGAACAGGACTT	
PlWRKY13qF	GAAACCCTCCTAACTTCTAC	Specific primer for qRT-PCR
PlWRKY13qR	AAACATTCATTCAACTCCC	
PlWRKY13F	ATGTTAAACCAGGCG	Full-length DNA and cDNA amplification
PlWRKY13R	CCAGAAGAAATTATT	
PlWRKY13(X)F	GCTCTAGAATGTTAAACCAGGCG	vector construction of GFP
PlWRKY13(K)R	GGGGTACCCCAGAAGAAATTATT	
PlWRKY13(E)F	CGGAATTCATGTTAAACCAGGCG	vector construction of VIGS
PlWRKY13(K)R	GGGGTACCATCAGCATCTCCATG	
pTRV1F	TTACAGGTTATTTGGGCTAG	
pTRV1R	CCGGGTTCAATTCCTTATC	Molecular detection of TRV
pTRV2F	TTTATGTTCAGGCGGTTCTTGTG	
pTRV2R	CAAACGCCGATCTCAAACAGTC	
PR1F	TACCCAGAGACGGTTCGACT	
PR1R	CACACGAGTTGGACCGGTAA	
PR2F	TGGCCAAAGGGGTCTCTAGA	
PR2R	TCCCATTTACGGCAAGCGTA	
PR4BF	ATCCCGCTCAACACTCTTGG	
PR4BR	TCCACAGAAAGCAGTCCACC	Primers for pathogenesis-related genes
PR5F	CAGTCTTCCCTCAGGCAAGG	
PR5R	GGTTTCACATGCGGGTTTCC	
PR10F	CCGGCAAGGATTTTCAAGGC	
PR10R	TTATCTTGATGGTCCCGGCG	

The underlined part, ′TCTAGA′, ′GAATTC′ and ′GGTACC′, are the added restriction enzyme sites of XbaI, EcoRI and KpnI, respectively. Primers of the pTRV1 vector were designed in RNA-dependent RNA polymerase elements (PCR product size was 647 bp in theory), and primers of the pTRV2 vector were designed in the MCS (PCR product size was 372 bp in theory).

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
