# Peer review of "PlWRKY13: A Transcription Factor Involved in Abiotic and Biotic Stress Responses in *Paeonia lactiflora"

_ijms, 2019, doi:10.3390/ijms20235953_

Round 1
Reviewer 1 Report
Authors responded to my questions and remarks. The responses are very interesting and clear; however, some points were not included into the manuscript (Comments 1 and 4).
I suppose that these points should be also discussed in the manuscript; thus, the minor revision of the manuscript is desirable.
Author Response
Point 1: Authors responded to my questions and remarks. The responses are very interesting and clear; however, some points were not included into the manuscript (Comments 1 and 4).
I suppose that these points should be also discussed in the manuscript; thus, the minor revision of the manuscript is desirable.
Response 1: Thank you for your suggestions. We have added the discussion of these two parts in the discussion section, please see Line 361-376, Page 15.
Reviewer 2 Report
The work from Wang and coauthors describes the characterization of the TF WRKY13 in Paeonia and its involvement in response to biotic and abiotic stresses. The work is clearly presented, and also the results are well presented, nevertheless there are some major concerns. The basic description of VIGS strategy is redundant and can be showed as supplementary figure, instead the results should show how the authors were able to avoid off target genes since the sequence homology of the WRKY genes is also well recognized. In this regard, I believe that VIGS cannot be the main proof of function for the TF and stable overexpression /knock out should be performed . Considering the standards of the IJMS, I suggest to add this experimental proof and resubmit the work.
Author Response
Point 1: The basic description of VIGS strategy is redundant and can be showed as supplementary figure, instead the results should show how the authors were able to avoid off target genes since the sequence homology of the WRKY genes is also well recognized.
Response 1: Thank you for your suggestions. In this article, we have listed the graphical representation of the vector used to build VIGS and simplified the description of VIGS in detail, see Line 475-476 and 491-495, Page 18. In addition, considering the sequence homology of the WRKY family, the non-conserved segment of PlWRKY13 gene (348 bp long sequence) was used as the target fragment to be inserted into the vector to avoid the interference of the homologous sequence of the WRKY family, which was supplemented in the results, see Line 182-183, Page 8. In barley, the same method was used to identify the functions of HvWRKY1 and HvWRKY2 in plant defense response (Ding et al., 2006).
Point 2: In this regard, I believe that VIGS cannot be the main proof of function for the TF and stable overexpression /knock out should be performed.
Response 2: Thank you for your Suggestions. Gene overexpression and knock out techniques can indeed prove the function of the target gene, but studies have shown that VIGS can effectively interfere with gene expression in a variety of plant systems, and it has been successfully used in many species, such as rice (Miki et al., 2005), barley (Christensen et al., 2004; Douchkov et al., 2005), wheat (Travella et al., 2006) and rose (Sui et al., 2018), etc.
References
Miki, D.; Itoh, R.; Shimamoto, K. RNA silencing of single and multiple members in a gene family of rice. Plant Physiol. 2005, 138, 1903-1913.
Christensen, A. B.; Thordal-Christensen, H.; Zimmermann, G.; Gjetting, T.; Lyngkjr, M. F.; Dudler, R.; Schweizer, P. The germinlike protein GLP4 exhibits superoxide dismutase activity and is an important component of quantitative resistance in wheat and barley. Mol. Plant-Microbe Interact. 2004, 17, 109-117.
Douchkov, D.; Nowara, D.; Zierold, U.; Schweizer, P. A high-throughput gene-silencing system for the functional assessment of defense-related genes in barley epidermal cells. MoL. Plant-Microbe Interact. 2005, 18, 755-761.
Travella, S.; Klimm, T. E.; Keller, B. RNA interference-based gene silencing as an efficient tool for functional genomics in hexaploid bread wheat. Plant Physiol. 2006, 142, 6-20.
Sui, X.; Zhao, M.; Xu, Z.; Zhao, L.; Han, X. RrGT2, A Key Gene Associated with Anthocyanin Biosynthesis in Rosa rugosa, Was Identifified Via Virus-Induced Gene Silencing and Overexpression. Int. J. Mol. Sci. 2018, 19, 4057.
Ding, X. D.; Schneider, W. L.; Chaluvadi, S. R.; Mian, M. A. R.; Nelson, R. S. Characterization of a Brome mosaic virus strain and its use as a vector for gene silencing in monocotyledonous hosts. Mol. Plant Microbe Interact. 2006, 19, 1229-1239.
Round 2
Reviewer 2 Report
In this revised version, the authors partially addressed my suggestions .I'm still not convinced that Vigs can be considered a sufficient proof of function . It's true that ten years ago Vigs represented a good method to interfere with the normal expression of the gene of interest, nevertheless as they also reported in the most recent paper IJMS in Rosa rugosa, a non model plant , a double check with a transient overexpression in tobacco or in a mutant plant line is usually required and then could be performed .
Author Response
Dear reviewer:
Thank you very much for your serious attitude and advice on this article. In the discussion section, we added a discussion on VIGS technology and pointed out its shortcomings in specific applications, as shown in Line 405-414, Page 16. We preliminarily studied the regulatory role of P. lactiflora WRKY in disease resistance, and the experiment still needs further exploration. Thanks again for your advice.
This manuscript is a resubmission of an earlier submission. The following is a list of the peer review reports and author responses from that submission.
Round 1
Reviewer 1 Report
The manuscript by Wang et al. is devoted to problem of participation of PlWRKY13 in responses of Paeonia lactiflora on stressors. The work seems to be interesting; however, there are some questions and comments.
The major question: it is known that propagating electrical signals can induce activation of plant defense genes (FEBS Lett., 1996, 390: 275-279; Nature, 2013, 500: 422-426); this activation is important part of the systemic plant adaptation response on local action of various abiotic and biotic stressors (Prog. Biophys. Mol. Biol., 2019, 146: 63-84). Using of the some factors in the manuscript was local: salt and waterlogging treatments influenced on roots; high and low temperatures influenced on leaves (maybe in different manner on different leaves) (P. 16, Section “4.5. Stress treatments and infection with A. tenuissima”). The changes in expression were measured in leaves. Thus, generation and propagation of electrical signals is very probable under these conditions (e.g. electrical signals can be induced by cold (Physiol. Plant., 2004, 120: 265-270; Front. Plant Sci., 2013, 4: article 239), heating and burning (Photosynth. Res., 2018, 136: 215–228; Funct. Plant Biol., 2019, 46: 328-338), watering (Plant Cell Environ., 2007, 30: 79–84; J. Plant Phys., 2018, 223: 32-36), etc). I suppose that this problem should be discussed: can electrical signals participate in changes of investigated defense genes?
Minor points:
1. P. 1, line 12: I suppose that functions of WRKY family and PlWRKY13 gene should be very briefly described in beginning of the Abstract. It can make text more clear for readers.
2. Figures 3C and 3F: dynamics of expression rate had several maximums; only one maximum was observed in Figures 3B, 3E and 3D. Was it connected with different mechanisms of the expression response? It should be clarified.
3. Figure 6A: Dynamics changes in JA and SA seems to be negatively correlated. Is it right? If yes, what are reasons of the effect?
4. P. 16, lines 408-409: Why 6 000 µmol m-2 s-1 was used as illumination? It is very high light intensity (light intensities under environmental conditions usually are less than 2 000 µmol m-2 s-1).
Thus I suppose that revision is necessary.
Reviewer 2 Report
The manuscript "PlWRKY13: A transcription factor involved in abiotic and biotic stress responses in Paeonia lactiflora" submitted to IJMS has been reviewed. It focused on the cloning and functional analyses of TF gene in peony.
My concerns are in the gene cloning and expression profiling parts as mentioned in the attachment. Other than that, the experiment was designed well and data collected support the conclusions. But the English language remains a professional polishing. Please see attachment for questions and comments.

Reviewer 3 Report
This study reports the results of a study on the expression of PlWRKY13 gene in Paeonia lactiflora. The manuscript is potentially interesting, fairly well reported and fits within the scope of the journal. I have some major concerns about the design of this study and its theoretical framework:
Aims are not clearly stated. Which hypothesis are you testing? How is the paragraph 2.1 going to answer your question? There is no clue how you did chose species for Fig 1D How is the paragraph 2.2 going to answer your question? Why do you want to know the cellular localization of PlWRKY13 protein? But, most importantly, why did you use N. benthamiana to run this test instead of Peonia? I do not see why you should report the results of semi-qPCR when you have the qPCR. Please drop any part associated with semi-qPCR. Results from data analyses are never reported. You have to always report F (or other statistic), df a P-values. Methods are not detailed in terms of number of samples and number of replicates. Also description of control groups is missing. How do you explain the fluctuating patterns in Fig 3C and 3F? In several experiments (e.g. Fig 3) Where the samples for gene expression collected at the same time of the day? Looking at Fig3B-E it looks like there is a patters. Maybe PlWRKY13 expression follows circadian rhythms? Discussion do not tell us which is the big novelty. How do your results impact the big picture?